evolution/genomics/palaeontology

phylogeny, evolutionary history, Cretaceous–Palaeogene mass extinction, Mesozoic, predation, Cretaceous Terrestrial Revolution

**Author for correspondence:**
Chenyang Cai
e-mail: cycai@nigpas.ac.cn

# Integrated phylogenomic and fossil evidence of stick and leaf insects (Phasmatodea) reveal a Permian–Triassic co-origination with insectivores

Erik Tihelka[1], Chenyang Cai[1,3], Mattia Giacomelli[1,2], Davide Pisani[1,2] and Philip C. J. Donoghue[1]

[1]School of Earth Sciences, and [2]School of Biological Sciences, University of Bristol, Life Sciences Building, Tyndall Avenue, Bristol BS8 1TQ, UK
[3]State Key Laboratory of Palaeobiology and Stratigraphy, Nanjing Institute of Geology and Palaeontology, and Centre for Excellence in Life and Paleoenvironment, Chinese Academy of Sciences, Nanjing 210008, People's Republic of China

 ET, 0000-0002-5048-5355; CC, 0000-0002-9283-8323; MG, 0000-0002-0554-3704; DP, 0000-0003-0949-6682; PCJD, 0000-0003-3116-7463

Stick and leaf insects (Phasmatodea) are a distinctive insect order whose members are characterized by mimicking various plant tissues such as twigs, foliage and bark. Unfortunately, the phylogenetic relationships among phasmatodean subfamilies and the timescale of their evolution remain uncertain. Recent molecular clock analyses have suggested a Cretaceous–Palaeogene origin of crown Phasmatodea and a subsequent Cenozoic radiation, contrasting with fossil evidence. Here, we analysed transcriptomic data from a broad diversity of phasmatodeans and, combined with the assembly of a new suite of fossil calibrations, we elucidate the evolutionary history of stick and leaf insects. Our results differ from recent studies in the position of the leaf insects (Phylliinae), which are recovered as sister to a clade comprising Clitumninae, Lancerocercata, Lonchodinae, Necrosciinae and *Xenophasmina*. We recover a Permian to Triassic origin of crown Phasmatodea coinciding with the radiation of early insectivorous pararetiles, amphibians and synapsids. Aschiphasmatinae and Neophasmatodea diverged in the Jurassic–Early Cretaceous. A second spur in origination occurred in the Late Cretaceous, coinciding with the Cretaceous Terrestrial Revolution, and was probably driven by visual predators such as stem birds (Enantiornithes) and the radiation of angiosperms.

# 1. Introduction

Stick and leaf insects (Phasmatodea) are a charismatic group of over 3100 herbivorous polyneopteran insects mostly occurring in the tropics and subtropics that are best known for mimicking plants [1–3]. Almost every aspect of their biology has been co-opted to increase their resemblance with plant foliage; their cryptic coloration and variously elongated or flattened body parts that resemble live and dead leaves, some species resemble moss and ferns, some look like bark, while others live hidden under stones [4] (figure 1). Many remain virtually motionless throughout the day [5], swaying in light wind just like the twigs and leaves they imitate [6]. Some species are known to perform startle displays consisting of leg and wing movements, and emit noxious secretions from their pygidial glands or even regurgitate their gut content, while others obtain additional protection from thorn-like projections on their body [7–11]. This impressive array of defensive morphologies and behaviours in Phasmatodea most likely originated as a response to visually hunting insectivorous predators; today, birds, lizards and spiders are considered to be the main predators of stick and leaf insects [4]. However, discriminating among competing evolutionary explanations for the remarkable suite of phasmatodean defensive adaptations that have shaped their modern diversity requires a reliable phylogeny and a suite of fossil evidence in order to calibrate phasmatodean evolution to geologic time. Neither of these pre-requisites have been met to date. Besides the fundamental split between the relictual genus *Timema* and the remaining stick insects, the Euphasmatodea, the phylogenetic relationships among phasmatodean subfamilies remain uncertain [1,3,12–15].

Given the fragile nature of phasmatodean bodies and the concealed nature of important diagnostic characters, the fossil record of these insects is sparse, and reliably assigned fossils are restricted only to a handful of deposits with exceptional preservation [16,17]. The interpretation of fossils assigned to Phasmatodea has stirred considerable controversy in the past decades [17–20]. Most recent molecular clock estimates have arrived at a Cretaceous to Palaeogene origin of stick and leaf insects [15,21–25], making them one of the youngest insect orders [21,26]. However, there are records of phasmatodeans extending deep into the Mesozoic [16,27,28], implying consistently earlier estimates of their origin [29]. Recent molecular clock analyses of Phasmatodea [15] appear also to have underestimated the age of webspinners, the sister group to Phasmatodea [21], which have records extending back to the Jurassic [30], as well as the age of other polyneopteran orders such as Mantophasmatodea [31] and Blattodea [5].

Resolving the timing of the evolution of Phasmatodea has implications for our understanding of the responses of insects to the Cretaceous–Palaeogene (K–Pg) mass extinction event. If stick and leaf insects originated in the Cretaceous and subsequently diversified in the Cenozoic [15,22,24], this implies a possible loss of early-diverging lineages during the K–Pg event. This would be a surprising but notable result, given that taxic analyses have consistently demonstrated that insect diversity was generally not affected by mass extinction events [32–36] and that the Cretaceous insect fauna included most orders and suborders familiar to entomologists today [37]. Moreover, the biology of stick and leaf insects also makes them excellent model organisms for understanding the macroevolutionary impacts of major radiations of insectivorous tetrapods including mammals and birds. If stick insects indeed diversified during the Palaeogene [15,22], this would indicate that their radiation may have been driven by the radiation of crown birds during this period [38]. However, insectivorous vertebrates [39–41] as well as plant-mimicking insects [42,43], already existed by the Permian.

Here, we address these questions surrounding the pattern and timescale of phasmatodean evolution. We use recently published transcriptomes for 38 phasmatodean species representing all major lineages from Simon *et al.* [15]. We re-analyse this dataset using methods accounting for compositional heterogeneity [44] to recover a well-supported backbone phylogeny of the order. We review the fossil record of Phasmatodea and select six calibration points representing the oldest reliable representatives of their linage, based on a combination of fossil, phylogenetic, stratigraphic, geochronological and biogeographic evidence, following the best practice recommendations [45] to revise the timing of phasmatodean diversification. We discuss methodological caveats associated with the selection of fossils for understanding the timescale of insect evolution.

# 2. Material and methods

## 2.1. Dataset generation

For phylogenetic reconstruction, we used a transcriptome dataset generated by Simon *et al.* [15], which includes 38 phasmatodean species belonging to all major clades. With an average of 2274 genes for each

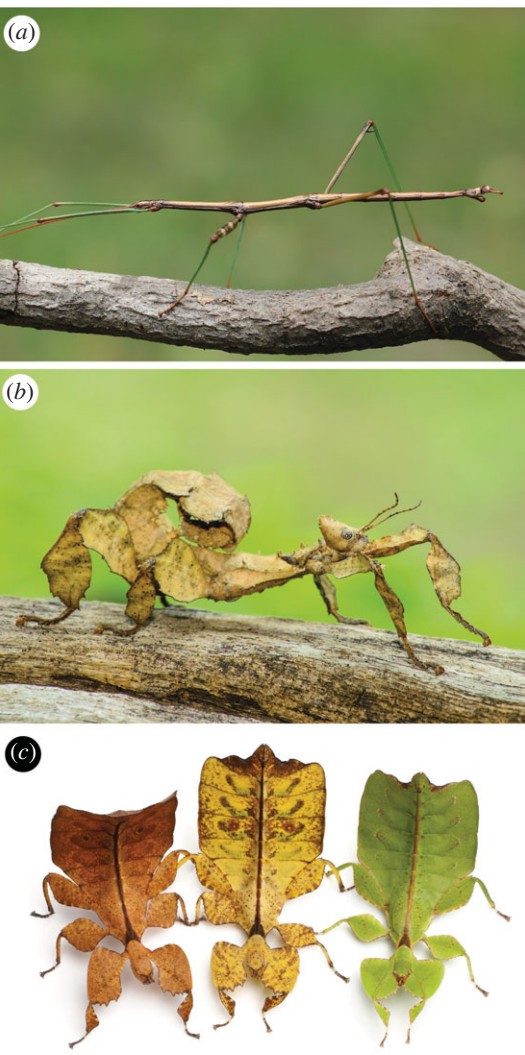

**Figure 1.** Extant diversity of stick and leaf insects (Phasmatodea). (a) *Diapheromera femorata* (Diapheromeridae), (b) *Extatosoma tiaratum* (Lancerocercata) and (c) *Phyllium westwoodii* (Phylliinae). Courtesy of Brian Lasenby (a), S. Aedka (b) and Eric Isselee (c).

sampled species, this represents the most comprehensive molecular dataset for Phasmatodea compiled to date. The original decisive amino acid alignment (AA$_{decisive}$) was downloaded from the publication's Dryad Digital Repository (doi:10.5061/dryad.65492qt).

To remove potentially misleading poorly aligned sequence regions and ease the convergence of runs, the original alignment was trimmed using data block mapping and gathering using entropy (BMGE v. 1.1, [46]) with BLOSUM62 with the threshold value set at 0.4. This resulted in a reduction of the dataset by approximately 62% from 387 987 to 148 945 sites. We tested the effects of outgroup selection on the recovered topology, since the inclusion of distant outgroups with highly divergent sequences relative to the ingroup can lead to the recovery of misleading topologies [47,48]. Moreover, distant outgroups reduce the precision of molecular clock estimates [49]. As such, we generated two datasets: one with 23 outgroup taxa spanning the Polyneoptera used in the original study [15], and a reduced dataset limited to four outgroups including the webspinners (Embioptera) and gladiators (Mantophasmatodea), the closest relatives of Phasmatodea [21].

## 2.2. Phylogenomic analyses

Phylogenetic analyses were conducted under the LG4X + R implemented in IQ-TREE and the infinite mixture model CAT-GTR + G in PhyloBayes. The LG4X + R analysis focused on the dataset with distantly related outgroups removed and was run in IQ-TREE v. 1.6.10 using 1000 ultra-fast bootstraps [50]. For the CAT-GTR + G runs, both datasets were analysed in PhyloBayes MPI 1.7. Two

independent Markov chain Monte Carlo (MCMC) chains were run until convergence (maxdiff < 0.3), the bpcomp program was used to generate output of the largest (maxdiff) and mean (meandiff) discrepancy observed across all bipartitions.

## 2.3. Molecular clock analyses

Six fossils were used to provide soft minimum and soft maximum constraints on six or five nodes of the phylogenomic dataset including Phasmatodea, Embioptera and Mantophasmatodea. The fossils were selected following best practice recommendations [45] and are listed in the electronic supplementary material.

Several major differences in calibration strategy compared with the previous calibration of Simon *et al.* [15] were made:

(i) The root of the tree was calibrated by the oldest crown mantophasmatodean *Juramantophasma sinica* from the Middle Jurassic Daohugou biota in China [31].
(ii) The node separating Embioptera and Phasmatodea was constrained with the Middle Jurassic stem-phasmatodean *Adjacivena rasnitsyni* from the same deposit [27]. The male specimen of the species possesses extensions of the 10th tergum, which have been interpreted as homologous with the vomer, a distinct apomorphy of Phasmatodea [51]. Moreover, affinity with stem Phasmatodea is supported by the ovipositor concealed by an operculum in female specimens [27]. Besides *Ad. rasnitsyni*, a further uncontested stem-stick insect from the Middle Jurassic Daohugou biota, *Aclistophasma echinulatum*, has been reported shortly before the submission of this paper [28]. Furthermore, the oldest webspinners are known from the same Middle Jurassic deposit [30].
(iii) The recently described *Tumefactipes prolongates* from mid-Cretaceous Burmese amber was used to calibrate stem Timematidae [52].
(iv) *Echinosomiscus primoticus* [26] is an enigmatic euphasmatodean fossil known from Burmese amber that has been used by Simon *et al.* [15] to calibrate the split between Timematodea and Euphasmatodea. However, the precise systematic position of the species is uncertain [26] so to test the robustness of the molecular clock analysis, we analysed two datasets, one including *E. primoticus* as a constraint on crown Lonchodidae and one without the fossil.
(v) To provide soft maximum age constraints for each calibrated node, we used the ages of well-explored insect Lagerstätten that lack representatives of the given lineages [53]. This eliminated the need to use arbitrarily chosen maximum constraints.

Molecular clock analyses were run in PAML 4.7 [54,55]. We obtained 200 000 trees with a sampling frequency of 50 and discarded 10 000 as burn-in. Default parameters were set as follows: 'cleandata = 0', 'BDparas = 1 1 0', 'kappa_gamma = 6 2', 'alpha_gamma = 1 1', 'rgene_gamma = 2 20', 'sigma2_gamma = 1 10' and 'finetune = 1: 0.1 0.1 0.1 0.01 0.5'. Convergence was assessed by plotting posterior mean times from the first run against the second run. Both the autocorrelated and independent rate relaxed-clock models were used.

Analyses were run using a uniform distribution between minimum and maximum node age constraints, reflecting the presumption of an equal probability of node age per unit time within this time interval. The interval was augmented by soft bounds with maximum and minimum tail probabilities of 2.5%. To test the robustness of our analyses to different model parameters, we used non-uniform truncated Cauchy distributions ($p = 0.1, 0.5, 0.9$) that differ in terms of the timing of the bulk of the prior probability density, reflecting prior beliefs that the fossil minima are good to poor reflection of the true time of divergence, respectively [56] (figure 3). Because two widely separated nodes in our molecular clock analyses, stem Timematidae and crown Lonchodidae, had the same minimum constraint (98.17 Ma), a uniform distribution was used for the latter alongside the Cauchy distributions on other nodes. We also explored the impact of using competing relaxed-clock models; the autocorrelated rate model assumes that rates are heritable while the versus independent rates model allows rates to vary between branches regardless of genealogy (figure 4).

# 3. Results

## 3.1. Phylogeny of Phasmatodea and the position of leaf insects (Phylliinae)

Regardless of the inference model used in our analyses with reduced taxon sampling, identical topologies were recovered (maxdiff = 0 for our PhyloBayes run performed on the reduced dataset [52], figure 2;

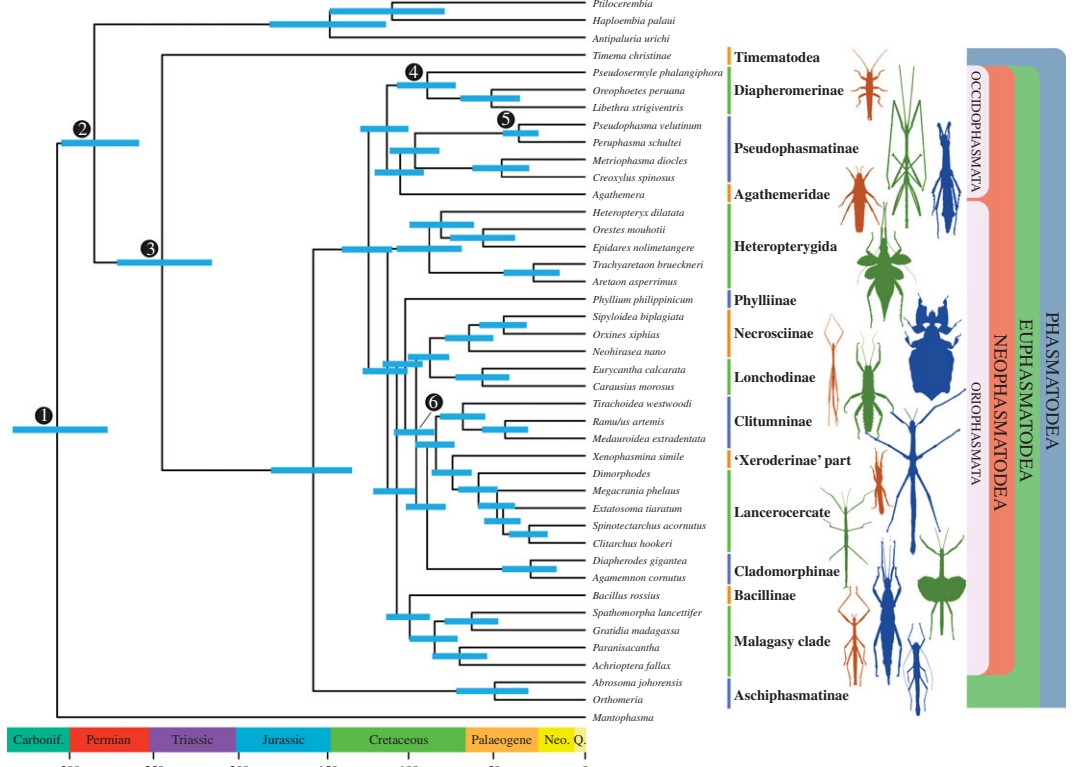

**Figure 2.** Dated phylogenetic tree of stick and leaf insects (Phasmatodea) based on a PhyloBayes re-analysis of the transcriptome dataset of Simon *et al*. [15], excluding distantly related outgroups. All nodes are fully supported (Bayesian posterior probability (BPP) = 1). Ages were estimated based on 190 000 separately generated trees constrained by six calibration points including *E. primoticus* and analysed under the independent clock model in MCMCtree with uniform prior distributions. Numbered nodes indicate calibrations: 1, *J. sinica*; 2, *A. rasnitsyni*; 3, *T. prolongates*; 4, *Clonistria* sp.; 5, *Eophasmodes oregonense*; 6, *E. primoticus*. Carbonif., Carboniferous; Neo., Neogene; Q., Quaternary.

**Table 1.** Ninety-five per cent HPD age estimates of divergences between major phasmatodean clades, comparing results obtained when the enigmatic Cretaceous fossil *E. primoticus* was included and excluded from the analyses. Molecular clock analyses were run under the independent clock model in MCMCtree with uniform prior distributions.

| nodes | calibrated with *E. primoticus* (Ma) | calibrated without *E. primoticus* (Ma) |
|---|---|---|
| Phasmatodea–Embioptera | 305–260 | 301–236 |
| Timematodea–Euphasmatodea | 272–217 | 249–190 |
| Aschiphasmatinae–Neophasmatodea | 183–136 | 184–124 |
| Occidophasmata–Oriophasmata | 141–112 | 150–78 |

maximum-likelihood bootstrap (MLB) ≥ 88 for all nodes for the IQ-TREE, electronic supplementary material, figure S2). PhyloBayes analyses of the dataset with complete taxon sampling did not converge (maxdiff = 1) even after a prolonged period of running. Nonetheless, the resultant tree of Phasmatodea was identical to those obtained under reduced outgroup sampling and highly supported (electronic supplementary material, figure S3).

There were two major differences between our topology and that of Simon *et al*. [15]. Notably, our PhyloBayes re-analysis of the complete dataset recovered Dermaptera and Plecoptera as sister groups (Dermoplecopterida), contrary to the results obtained by Simon *et al*. [15] and recent insect phylogenies [21,57]. Both orders share the same number of tarsomeres, indistinguishable male gonopods, and reduced ovipositors [58] and the clade Dermoplecopterida has been recovered by analyses of nuclear ribosomal RNA [59,60] and an analysis of five nuclear genes and 13 mitochondrial protein-encoding genes [61]. However, because not all of our analyses converged and our taxon sampling of outgroups was slim, definitive conclusions are difficult to draw at this point. The second major difference with reference to the topology of Simon *et al*. [15] is the position of the leaf insects (Phylliinae). We recovered Phylliinae as sister

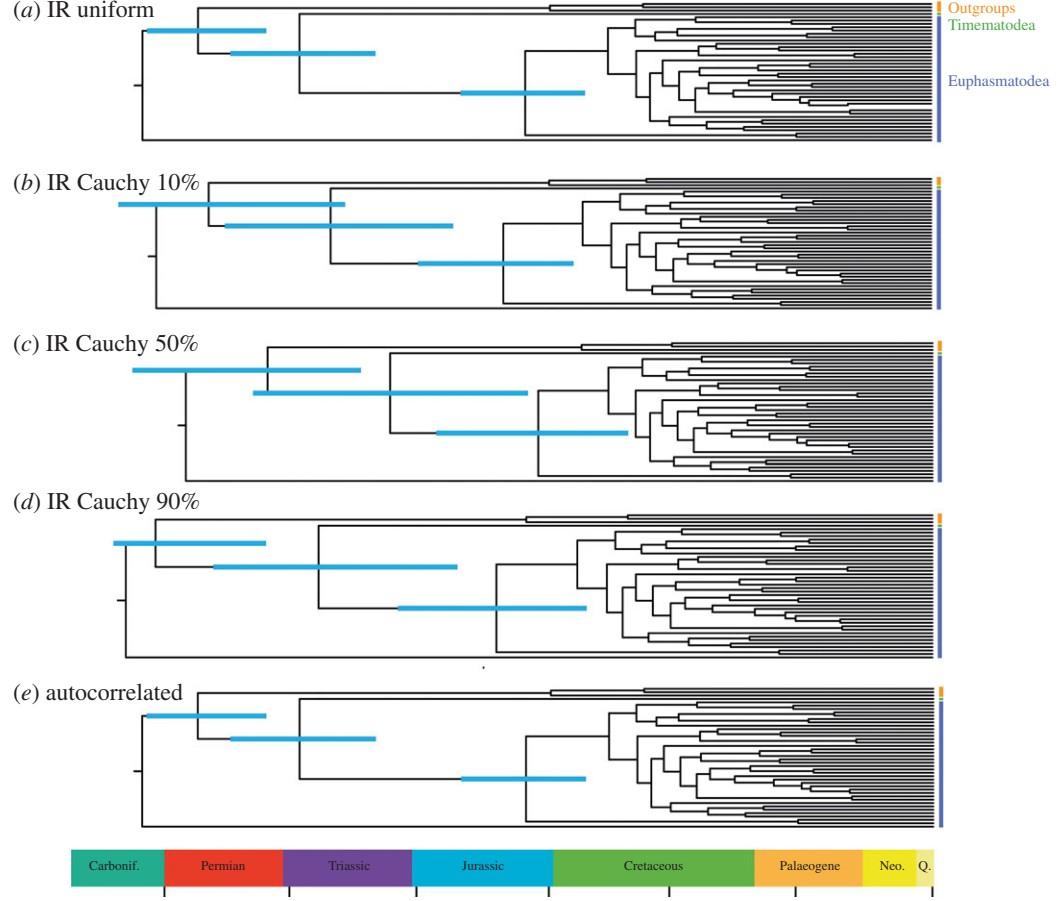

**Figure 3.** Posterior time estimates under different parameters. (*a*) Posterior time estimates when using a uniform calibration density prior distribution analysed under the independent clock model in MCMCtree, reflecting the conservative assumption that divergences may have occurred with an equal probability between the provided maximum and minimum fossil priors; (*b*) Cauchy 10% maximum calibration density prior distribution under the independent clock model in MCMCtree, reflecting the assumption that the fossil prior is a good approximation of the divergence date; (*c*) Cauchy 50% maximum calibration density prior distribution under the independent clock model in MCMCtree; (*d*) Cauchy 90% maximum calibration density prior distribution under the independent clock model in MCMCtree, reflecting the assumption that the fossil prior is a poor approximation of the divergence date; (*e*) posterior time estimates when using a uniform calibration density prior distribution analysed under the autocorrelated clock model in MCMCtree. All analyses were run with six fossil calibrations, including *E. primoticus*. Carbonif., Carboniferous; Neo., Neogene; Q., Quaternary.

to a clade comprising Clitumninae, Lancerocercata, Lonchodinae, Necrosciinae and *Xenophasmina* with full support (BPP = 1). Simon *et al.* [15] recovered leaf insects as sister to Bacillinae with a non-parametric bootstrapping support of 96. Bacillinae is now recovered as sister to the Malagasy clade of stick insects. The placement of leaf insects has been inconsistent in past studies of Phasmatodea [22,23,62], but the presently recovered topology is more consistent with the morphology-based classification of Key [63], who suggested expanding Phylliinae to include eight subfamilies including Necrosciinae.

## 3.2. The timescale of stick and leaf insect evolution

Regardless of whether *E. primoticus* was used as a minimum constraint for crown Lonchodidae or not, both molecular clock analyses yielded overlapping dates (table 1 and figure 2; electronic supplementary material, figure S4). According to our results, the split between *Timema* and Euphasmatodea occurred around the Middle or Late Triassic, although the confidence intervals obtained are quite wide and also include the Permian and Early Jurassic (95% confidence interval (CI) with *E. primoticus*: 272–217 Ma; CI without *E. primoticus:* 249–190 Ma). Nonetheless, this represents an older date than estimated by most recent molecular clock analyses, which have agreed on a Cretaceous–Palaeogene origin of Phasmatodea [15,22,24].

The split between Aschiphasmatinae and Neophasmatodea was estimated to have occurred around the Late Jurassic averaging 154 and 158 Ma in the two estimates (CI with *E. primoticus*: 183–136 Ma; CI

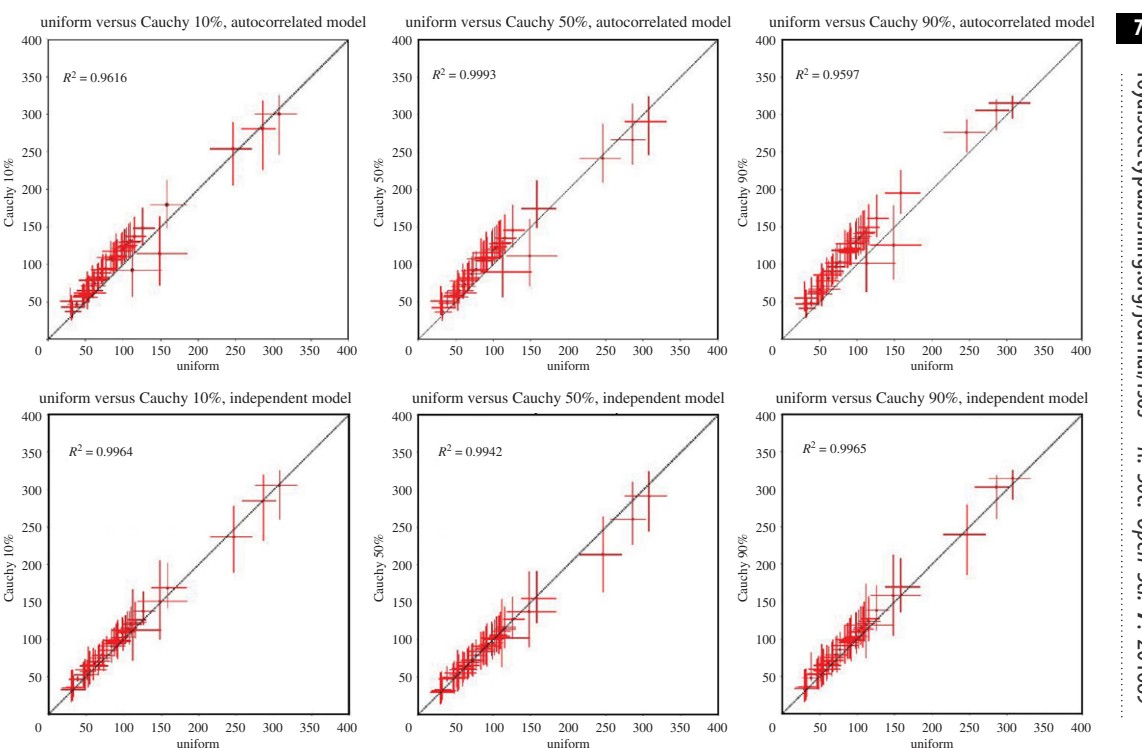

**Figure 4.** Scatterplot of the posterior divergence times derived from the analyses of uniform prior distribution against the ones from Cauchy distributions. Top row: autocorrelated model; bottom: independent model. Dots represent the posterior mean times; red lines represent the HPD intervals.

without *E. primoticus*: 184–124 Ma). Both analyses converged on an approximately Aptian origin of Occidophasmata and Oriophasmata averaging 126 and 125 Ma (CI with *E. primoticus*: 141–112 Ma; CI without *E. primoticus*: 150–78 Ma). Occidophasmata and Oriophasmata subsequently diversified in the Late Cretaceous and Palaeogene.

Our molecular clock estimates are obviously compatible with the age of phasmatodean fossils used in calibration; however, this evolutionary timescale is also congruent with the broader fossil record of phasmatodeans, which was not be used in calibration. These include undescribed Aptian–Albian euphasmatodean eggs [26], undescribed euphasmatodean nymphs and adults from the mid-Cretaceous [64,65], the Eocene leaf insects *Eophyllium messelensis* from the Messel Pit in Germany [66] and undescribed putative members of Euphasmatodea in Eocene Baltic amber [67]. We explored whether this timescale was robust to methodological assumptions, including the degree to which the fossil clade age minima approximate the true clade age. If the phylogenetic signal present in the dataset is strong, then the selection of priors would be expected to have little effect on divergence time estimates. On the other hand, if the data are less informative, posterior estimates would be expected to vary with the selection of different priors [68]. We found that regardless of the clock models or the calibration strategies used, we obtained very similar divergence dates (figures 3 and 4; electronic supplementary material, figures S4–S12). Figure 4 shows strong correlation ($R^2 \geq 0.9597$) between posterior divergence times derived from the analyses of uniform prior distributions against the ones from Cauchy distributions, indicating that the divergence date results obtained are robust and vary little under different model assumptions. Despite their general overlap, divergence time estimates showed limited variation with different model assumptions. For example, Cauchy 90% maximum calibration density prior distribution resulted in earlier median divergence times (figure 3*d*), reflecting the model's assumption that the fossil prior is a poor approximation of the divergence date.

# 4. Discussion

## 4.1. Sources of incongruence in divergence estimates of Phasmatodea

Sandoval *et al.* [25] concluded that the genus *Timema* diverged at only 20 Ma, but their molecular clock analysis used no fossil constraints and assumed that the genus originated in southern USA or northern

Mexico where it was at most as old as the local mountain ranges. However, fossil timematids have been recorded from the Cretaceous of northern Myanmar [52] rendering this assumption unlikely. Davis *et al.* [29] arrived at a Permian (*ca* 290 Ma) origin of Phasmatodea based on a supertree approach and assuming that Orthoptera is the sister group of stick insects, which is, however, not supported by more recent analyses (e.g. [57]). Aside from these early studies, all subsequent investigations have consistently recovered a Cretaceous to Palaeogene split between *Timema* and Euphasmatodea, marking the diversification of crown stick and leaf insects [15,22–24]. All of these studies used the Cretaceous *Renphasma sinica* from the Early Cretaceous Yixian Formation in Liaoning, China [16], or undescribed Cretaceous phasmatodean eggs [69] as the oldest unequivocal representatives of the order alongside one to four other calibrations. The significantly older node ages obtained by us result in large part from using the Middle Jurassic *Adjacivena rasnitsyni*, whose phasmatodean affinity is supported by otherwise rarely preserved genital structures [27], to establish the minimum constraint on the age of total-group Phasmatodea. Further support for this critical calibration is found in *Aclistophasma echinulatum*, from the same deposit, whose stem-phasmatodean affinity has been established through formal phylogenetic analysis [28]. Furthermore, the oldest uncontested stem-webspinners, which would be used to calibrate the same node, are of the same age [30]. Unlike prior studies, our calibrations have explicit and informed maxima, not merely minima, defined based on the age of fossil deposits that have yielded diverse and well-explored fauna but do not include representative members of the clade being calibrated [53]. This approach obviates the use of arbitrary maxima, uniformed choice among which is known to have an undue and significant impact on divergence time estimates [56,70,71].

Further refinement of phasmatodean molecular ages will necessitate the inclusion of more calibration points [72,73]. Throughout the twentieth century, it has been assumed that the fossil record of stick and leaf insects goes as far back as the Permian [74], but restudy has cast doubt on their purported phasmatodean affinity [17–20]. Early stick insects are readily confused with diverse orthopterans and phylogenetic affinities are most convincingly demonstrated based on genital characters which rarely preserve [18]. As such, unequivocal fossil stick and leaf insects are known almost exclusively from localities with exceptional preservation, such as amber deposits. The low numerical abundance of phasmatodean fossils in Lagerstätten is not surprising, given their arboreal lifestyle and limited mobility which results in a lower fossilization potential. Phasmatodeans are rare in flight traps placed in tropical rainforests and it has been estimated that they constitute less than 0.06% of the total insect biomass in the tropics [75]. The unsurprising scarcity of unambiguous pre-Cretaceous stick and leaf insects will probably continue to be a major reason for the broad uncertainties in the estimates of the age of this group in future studies. Another factor affecting the estimated divergence ages is the breadth of taxon sampling, since divergence dates essentially represent the ages of the extant clade circumscribed by the sampled taxa, not the ages of clades *per se*. Thus, inclusion of more diverse Phasmatodean taxa would be expected to alter the ages of shallower nodes. However, these changes are difficult to foresee at present because of remaining uncertainty regarding the classification of Phasmatodea below the level of subfamilies.

## 4.2. Drivers of stick and leaf insect diversification

A recently described stem-stick insect from the Middle Jurassic Jiulongshan Formation in China exhibits abdominal extensions and femoral spines [28], providing evidence that the earliest fossil representatives of the order possessed morphological adaptations for mimicking plants. Plant mimicry is universal throughout extant crown Phasmatodea and is the defining feature of the order [4,12], implying that the last common ancestor of crown Phasmatodea was probably already involved in mimicking plants as well. Ecological theory predicts that the origin and maintenance of mimicry is primarily driven by at least two selective pressures: (i) predation, and (ii) the morphology of the models being mimicked [76–78].

It is unlikely that invertebrate predators such as arachnids played a major role in phasmatodean diversification as their vision was probably limited [79]. Visually hunting predators, namely synapsids and sauropsids, became very diverse in the Middle Permian [80]. Some of the oldest members of these tetrapod clades possess dentition characteristic of an insect-based diet, namely sharp, peg-like teeth that probably enabled them to seize small insects and swallow them whole [81–83]. The earliest direct evidence of insectivory come from the oral content of Lower Permian (Artinskian) parareptiles [41]. Mammaliaforms diversified in the Late Triassic and Early Jurassic, as indicated by molecular clock analyses and fossil evidence [40,84–86]. Dentition and jaw morphology suggest that these mammaliaforms and early mammals depended on a predominantly insect-based diet [39,87–91]. The earliest diverging pterosaur clade, the Late Triassic Eopterosauria, were insectivorous as

suggested by dental evidence, and further insectivorous pterosaurs are known from the Jurassic to the Early Cretaceous [92,93]. Prominent Cretaceous insectivores were the Enantiornithes (opposite birds) which were probably restricted to soft-bodied arthropods, as indicated by the absence of gizzard stones which are used by extant birds feeding on insects with a tough exoskeleton [94,95]. Insectivores that diversified after the K–Pg extinction event that decimated non-avian dinosaurs included the amphibians, namely frogs, and modern birds [96,97].

While the precise date of the origin of flowering plants remains contentious, the earliest unequivocal fossil angiosperms date back to the Early Cretaceous, when gymnosperms were the most diverse group of land plants [98]. Angiosperms rose to prominence in the mid-Cretaceous and came to dominate forests by the Campanian–Maastrichtian (84–65 Ma), displacing the gymnosperms [99,100]. The radiation of angiosperms was associated with an explosive co-diversification of pollinators, herbivores and predators [101,102], in a period of biotic turnover known as the Cretaceous Terrestrial Revolution [102].

We have shown that when the oldest representatives for each lineage are used to calibrate the phylogeny of Phasmatodea, the age of crown stick insects is pushed back into the late Palaeozoic to early Mesozoic. The CI of crown Phasmatodea diversification covers the Permian to Jurassic. This coincides with the appearance of gliding and terrestrial vertebrates such as amphibians, synapsids and sauropsids in the fossil record that apparently became very diverse in the Middle Permian [80]. The origination of Phasmatodea also overlaps with the early diversification of insectivorous mammaliaformes; since some Jurassic mammals were arboreal [40,103], phasmatodeans may have been among their prey. The CI of crown Phasmatodea falls outside of the range of Late Triassic to Early Jurassic insectivorous pterosaurs, although some overlap is present under the independent rates model with Cauchy 50% maximum calibration density prior distribution (figure 3). The second spur of phasmatodean diversification during the Late Cretaceous coincides with the diversification of Enantiornithes, which were mostly arboreal and insectivorous [104,105], making them likely predators of stick insects. Enantiornithes and stick insects co-occurred in the same habitats in Mesozoic rainforests, as suggested by the fact that both putatively insectivorous Enantiornithes and phasmatodeans have been recovered from Cretaceous Burmese amber [26,106,107]. The Late Mesozoic phasmatodean radiation also overlaps with the Cretaceous Terrestrial Revolution. The majority of extant phasmatodeans feed on and mimic angiosperms, although the host range of *Timema* also includes conifers such as cypresses and pines [18,108]. Other factors that may have contributed to the diversification of Phasmatodea were the acquisition of pectinase genes from gut microbes [109] and the origin of novel oviposition strategies [25], but these hypotheses will have to be re-evaluated in the light of the new phasmatodean backbone phylogeny proposed by Simon *et al.* [15] and herein.

The present molecular clock estimates suggest that phasmatodeans may have continued to diversify across the K–Pg boundary. That stick insects have not suffered a major extinction in this period is corroborated by the fact that the extinct Susumaniidae that date back to the Jurassic crossed the K–Pg boundary and have been reported from Eocene deposits [67,110]. The extant taxa Timematodea and Euphasmatodea likewise crossed the K–Pg boundary [26,52].

# 5. Conclusion

Stick and leaf insects represent a remarkable model system for studying the evolution of mimesis and understanding the macroevolutionary impacts of the diversification of insectivores during the Mesozoic. However, molecular phylogenies have yielded strongly divergent topologies and suggested a Cretaceous–Palaeogene origin of the group [15,22–24], making it potentially one of the youngest insect orders [21,26]. Such a late diversification of the stick insects would be surprising since the fossil record of insectivores [39,40] and plant-mimicking insects spans the entire Mesozoic [42,43]. We re-analysed a transcriptome dataset for Phasmatodea recently sequenced by Simon *et al.* [15] to elucidate the backbone phylogeny of the order and provide a revised timescale of phasmatodean evolution. Using methods that account for compositional heterogeneity of molecular datasets, we recovered leaf insects (Phylliinae) as sister to a clade including Clitumninae, Lancerocercata, Lonchodinae, Necrosciinae and *Xenophasmina*. We used a comprehensive set of calibration points selected according to best practice recommendations [45], including the earliest representatives of all studied clades and realistically constrained soft upper bounds based on data from localities with exceptional preservation. We recovered a Permian–Triassic radiation of crown Phasmatodea. This period has been marked by the origin and subsequent diversification of some of the earliest gliding, terrestrial and arboreal insectivores [41] that may have represented a key selection pressure on early stick insects. Subsequent diversification of stick and leaf insects has coincided with the diversification of stem birds, frogs and

angiosperms. Our backbone phylogeny of Phasmatodea will be useful for testing further hypotheses about the evolution of this iconic insect order.

Data accessibility. Analysed files and outputs of molecular clock analyses are available from Mendeley Data: http://dx.doi.org/10.17632/z4z639sknw.3.

Authors' contributions. C.C. and E.T. conceived and designed the study, C.C. carried out the phylogenomic analyses, E.T. and M.G. carried out the molecular clock analyses, E.T. drafted the manuscript to which C.C., M.G., D.P. and P.C.J.D. contributed. All authors participated in the interpretation of the results and agreed on the final form of the paper.

Competing interests. The authors declare that they have no financial or non-financial competing interests.

Funding. Financial support was provided by the Strategic Priority Research Program of the Chinese Academy of Sciences (XDB26000000), the National Natural Science Foundation of China (41688103), the Second Tibetan Plateau Scientific Expedition and Research (2019QZKK0706) and a Newton International Fellowship from the Royal Society. M.G. and D.P. were supported by the European Union Horizon 2020 research and innovation programme under the Marie Skłodowska-Curie grant agreement (764840).

Acknowledgements. We are grateful to two anonymous reviewers for their constructive criticism. E.T. is indebted to Dr Jakob Vinther (University of Bristol) for valuable comments on insect predators in the Mesozoic.

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
