## [Reviewer comments · Royal Society Open Science]

Review History

Decision letter (RSOS-201689.R0)

Dear Mr Tihelka:

It is a pleasure to accept your manuscript entitled "Integrated phylogenomic and fossil evidence of stick and leaf insects (Phasmatodea) reveal a Permian-Triassic co-origination with insectivores" in its current form for publication in Royal Society Open Science. The comments of the reviewer(s) who reviewed your manuscript are included at the foot of this letter.

Best regards,

on behalf of Dr Maximilian Telford (Associate Editor) and Professor Kevin Padian (Subject Editor).

Associate Editor Dr Maximilian Telford Comments to Author:

Seems like a relatively small addition to the literature. The paper reanalyses an existing data set with some new fossil calibrations in a new attempt at dating the radiation of the stick insects. The paper goes on and slightly over interprets these results. Nevertheless an interesting group of insects and a new proposal for the date of their radiation bringing some new insights.

The authors have responded adequately to two decent reviews. They still need to attend to careless typos in the new version of the manuscript. Especially in the methods. Most of the section "Drivers of stick and leaf insect diversification" seems to be story telling and I suggest that this is considerably toned down.
